# Synthesis and Anticancer Activity of Indole-Functionalized Derivatives of Betulin

**DOI:** 10.3390/pharmaceutics14112372

**Published:** 2022-11-04

**Authors:** Zuzanna Rzepka, Ewa Bębenek, Elwira Chrobak, Dorota Wrześniok

**Affiliations:** 1Department of Pharmaceutical Chemistry, Faculty of Pharmaceutical Sciences in Sosnowiec, Medical University of Silesia in Katowice, 4 Jagiellońska Str., 41-200 Sosnowiec, Poland; 2Department of Organic Chemistry, Faculty of Pharmaceutical Sciences in Sosnowiec, Medical University of Silesia in Katowice, 4 Jagiellońska Str., 41-200 Sosnowiec, Poland

**Keywords:** triterpenes, betulin, indole, anticancer activity

## Abstract

Pentacyclic triterpenes, including betulin, are widespread natural products with various pharmacological effects. These compounds are the starting material for the synthesis of substances with promising anticancer activity. The chemical modification of the betulin scaffold that was carried out as part of the research consisted of introducing the indole moiety at the C-28 position. The synthesized new 28-indole-betulin derivatives were evaluated for anticancer activity against seven human cancer lines (A549, MDA-MB-231, MCF-7, DLD-1, HT-29, A375, and C32). It was observed that MCF-7 breast cancer cells were most sensitive to the action of the 28-indole-betulin derivatives. The study shows that the lup-20(29)-ene-3-ol-28-yl 2-(1*H*-indol-3-yl)acetate caused the MCF-7 cells to arrest in the G1 phase, preventing the cells from entering the S phase. The performed cytometric analysis of DNA fragmentation indicates that the mechanism of EB355A action on the MCF-7 cell line is related to the induction of apoptosis. An in silico ADMET profile analysis of EB355A and EB365 showed that both compounds are bioactive molecules characterized by good intestinal absorption. In addition, the in silico studies indicate that the 28-indole-betulin derivatives are substances of relatively low toxicity.

## 1. Introduction

Cancer, as one of the leading causes of death and an important barrier to increasing life expectancy, has a major impact on society across the world. There were an estimated 19.3 million new cases of cancer and almost 10.0 million cancer deaths worldwide in 2020 [1]. Despite the significant development of new therapies, the resistance of cancer cells and serious side effects of the drugs used are still the most important challenges for medicine. Thus, there is a continuous need to develop new anti-cancer drugs. Natural products from plants have served as a primary source of oncological medications and continue to provide new plant-derived anticancer agents [2].

Pentacyclic triterpenes are a chemically diverse group of secondary metabolites of plant origin. Their representative is betulin (Figure 1) found in more than 200 different types of plants. It is mainly isolated from the outer bark of birch trees (*Betula*, *Betulaceae* family) and possesses a wide range of biological activities, including anticancer, anti-inflammatory, antiviral, and antimicrobial properties [3,4,5,6,7,8]. The promising anti-tumor activity of betulin has been evidenced both in cell culture models [9,10,11] and animal models [12,13,14]. Moreover, the structure of betulin molecules enables chemical modifications within the functional groups at the C-3, C-28, C-20, C-30, and C-17 positions and, thus, obtaining derivatives with optimized anticancer and pharmacokinetic properties [15,16,17,18,19,20].

Indole (1*H*-benzo[b]pyrrole, Figure 2) is one of the most widespread heterocyclic moieties in natural and synthetic bioactive compounds. A unique feature of the indole scaffold is its ability to interact with more than one receptor, which determines a variety of biological activities. Therefore, indole derivatives have a wide range of biological effects and are currently used in the treatment of various diseases [21,22].

The indolin-2-one scaffold is present in the structure of suntinib (anticancer drug) (Figure 2), which inhibits the Fms-like tyrosine kinase-3 receptor (FLT3). This multi-directional tyrosine kinase inhibitor has an inhibitory effect on vascular endothelial growth factor VEGF, platelet-derived growth factor (PDGF), and c-KIT receptor kinases. Sunitinib is used to treat various types of cancer, such as renal cell carcinoma (RCC), gastrointestinal stromal tumor (GIST), pancreatic neuroendocrine tumors (pNET), non-small cell lung cancer (NSCLC), and thyroid cancer [23,24].

Drugs that contain a chemical indole moiety and are clinically applicable include zafirlukast (anti-asthma), pindolol (anti-hypertensive), indomethacin (anti-inflammatory), roxindole (anti-psychotic), sumatriptan (anti-migraine), and arbidol (antiviral) (Figure 2) [21,22,25,26,27,28].

The indole scaffold has been a highly beneficial motif for the development of agents effective against cancer, including the drug-resistant types [22,29]. The introduction of an indole moiety to the structure of triterpene derivatives in order to obtain new compounds with anticancer activity mainly concerns the modification at the A-ring [30,31,32]. Significant anticancer activity was observed for the betulinic acid derivatives substituted with indolyl at positions C-2 and C-3. The obtained compounds were tested against pancreatic (MIAPaCa2), lung (A-549), ovary (PA-1), breast (HBL100), prostate (DU145), leukemia (K562), and colon (SW620) cancer cells. It was observed that the presence of a chlorine group in the indole ring at the C-5 position led to an increase in activity against MIAPaCa, PA-1, and SW620 cells (EC_50_ = 2.44–2.70 µg/mL) [30].

The aim of the study was to synthesize 28-indole-betulin derivatives. In the next step, the potential antitumor activity of the obtained compounds was investigated using in vitro models of lung cancer, triple-negative breast cancer, estrogen-receptor-positive breast cancer, colorectal adenocarcinoma, and melanoma. The impact on normal human cells was also evaluated. Then, the more effective compound was analyzed to determine whether the observed anti-cancer effect was due to the induction of cell death and/or the disturbance of cell cycle progression. In addition, ADMET analysis was performed, which is an important method for characterizing new chemical compounds as drug candidates.

## 2. Materials and Methods

### 2.1. Chemistry

The melting points were measured on an Electrothermal IA 9300 apparatus (Bibby Scientific Limited, Stone, Southampton, UK), and were uncorrected. HRMS (APCI) spectra were recorded on a Bruker Impact II instrument (Bruker). ^1^H and ^13^C NMR spectra were recorded in CDCl_3_ on a Bruker Avance III 600 spectrometer (Bruker, Billerica, MA, USA). The reactions and purity of the products were monitored by TLC (silica gel 60 F_254_ plates, Merck, Darmstadt, Germany). The plates were visualized by spraying with an ethanolic solution of sulfuric acid and heating to 100 °C. The purification of both compounds was carried out by column chromatography (silica gel 60 (0.063–0.200 mm); chloroform/ethanol 40:1, *v*/*v*). The reagents and solvents applied in the synthesis and chromatography were purchased from Merck (Darmstadt, Germany).

### 2.2. Synthesis of Indole Betulin Derivatives EB355A and EB365

A reaction flask containing 1 mmol of substrate (betulin or 3-acetylbetulin) and 5 mL of dichloromethane (CH_2_Cl_2_) was placed in an ice-water bath on a magnetic stirrer. 3-indole acetic acid (1.10 mmol) was added to the reaction mixture cooled to −10 °C, and a solution containing 1.12 mmol DCC and 0.08 mmol DMAP in 1 mL dichloromethane was gradually added dropwise. The cooling bath was then removed, and stirring was continued at room temperature for 24 h. In the next step of the synthesis, the reaction mixture was filtered to remove a by-product (*N*,*N*′-dicyclohexylurea). Dichloromethane was distilled from the filtrate using a rotary evaporator. The obtained products of the esterification reaction were purified by column chromatography (SiO_2_, chloroform/ethanol, 40:1, *v*/*v*).

*Lup-20(29)-ene-3-ol-28-yl 2-(1H-indol-3-yl)acetate* EB355A Yield 58%; m.p. 131–132 °C; R_f_ = 0.28 (chloroform/ethanol, 40:1, *v*/*v*); ^1^H NMR (CDCl_3_, 600 MHz) δ 0.78 (s, 3H, CH_3_), 0.83 (s, 3H, CH_3_), 0.96 (s, 3H, CH_3_), 0.98 (s, 3H, CH_3_), 1.00 (s, 3H, CH_3_), 1.68 (s, 3H, CH_3_), 2.44 (m, 1H, H-19), 3.21 (m, 1H, H-3), 3.82 (s, 2H, O=CCH_2_), 3.90 (d, 1H, *J* = 11.4 Hz, H-28), 4.32 (d, 1H, *J* = 10.8 Hz, H-28), 4.59 (s, 1H, H-29), 4.68 (s, 1H, H-29), 7.14–7.64 (m, 5H, indole), 8.10 (s, 1H, N-H, indole) (Appendix A); ^13^C NMR (CDCl_3_, 150 MHz) δ 14.75, 15.37, 16.02, 16.10, 18.28, 19.12, 20.77, 25.18, 27.04, 27.40, 27.99, 29.60, 29.77, 31.46, 34.15, 34.58, 37.14, 37.57, 38.70, 38.87, 40.85, 42.68, 46.44, 47.70, 48.81, 50.36, 55.29, 63.18, 79.00, 108.77, 109.81, 111.12, 118.90, 119.66, 122.22, 122.95, 127.27, 136.07, 150.21, 172.53 (Appendix A). HRMS (APCI) *m*/*z*: calcd for C_40_H_57_NO_3_ [(M-H)^−^]: 598.4260; found 598.4228 (Appendix A).

*3-Acetyl-lup-20(29)-ene-28-yl 2-(1H-indol-3-yl)acetate* EB365 Yield 52%; m.p. 128–130 °C; R_f_ = 0.54 (chloroform/ethanol, 40:1, *v*/*v*); ^1^H NMR (CDCl_3_, 600 MHz) δ 0.77 (br s, 9H, 3 × CH_3_), 0.87 (s, 3H, CH_3_), 0.92 (s, 3H, CH_3_), 1.60 (s, 3H, CH_3_), 1.97 (s, 3H, CH_3_C=O), 2.34 (m, 1H, H-19), 3.73 (s, 2H, CH_2_C=O), 3.80 (d, 1H, *J* = 10.8 Hz, H-28), 4.23 (d, 1H, *J* = 10.8 Hz, H-28), 4.39 (m, 1H, H-3), 4.50 (s, 1H, H-29), 4.59 (s, 1H, H-29), 7.00–7.55 (m, 5H, indole), 8.00 (s, 1H, N-H, indole) (Appendix A); ^13^C NMR (CDCl_3_, 150 MHz) δ 14.71, 16.02, 16.16, 16.50, 18.16, 19.10, 20.78, 21.36, 23.70, 25.13, 27.03, 27.95, 29.58, 29.76, 31.46, 34.08, 34.58, 37.05, 37.55, 37.80, 38.37, 40.86, 42.67, 46.43, 47.72, 48.80, 50.26, 55.37, 63.18, 80.95, 108.78, 109.85, 111.06, 111.12, 118.19, 119.66, 122.23, 122.79, 122.94, 127.27, 136.07, 150.20, 171.08, 172.53 (Appendix A); HRMS (APCI) *m*/*z*: calcd for C_42_H_58_NO_4_ [(M-H)^−^]: 640.4366; found 640.4359 (Appendix A).

### 2.3. Cell Culture and the Treatment

Normal human fibroblasts were purchased from Sigma Aldrich Inc. (St. Louis, MO, USA) and cultured in the all-in-one ready-to-use Fibroblast Growth Medium (Sigma Aldrich Inc., St. Louis, MO, USA). Human lung carcinoma cells A549 (CCL-185), human breast cancer cells MDA-MB-231 (HTB-26) and MCF-7 (HTB-22), human colorectal adenocarcinoma cells DLD-1 (CCL-221) and HT-29 (HTB-38), as well as human skin amelanotic melanoma cells A375 (CRL-1619) and C32 (CRL-1585), were acquired from the American Type Culture Collection (Manassas, VA, USA). HT-29 cells were cultured in McCoy’s 5A medium (Sigma Aldrich Inc., St. Louis, MO, USA), and DLD-1 cells in RPMI 1640 medium (PAN-Biotech, Aidenbach, Germany). The remaining cell lines were cultured in DMEM (Thermo Fisher Scientific, Waltham, MA, USA). Apart from the Fibroblast Growth Medium, all media were supplemented with inactivated fetal bovine serum (Gibco, Waltham, MA, USA) to a final concentration of 10%, as well as with the antibiotics: penicillin (100 U/mL), neomycin (10 µg/mL), and amphotericin B (0.25 µg/mL), which were obtained from Sigma Aldrich Inc. (St. Louis, MO, USA). Cells were cultured at 37 °C in a humidified atmosphere with 5% CO_2_. Sub-confluent cells were harvested with 0.25% trypsin-EDTA (Thermo Fisher Scientific, Waltham, MA, USA). The obtained compounds EB365 and EB355A were dissolved in DMSO (Sigma Aldrich Inc., St. Louis, MO, USA) to prepare stock solutions (10 mg/mL). For the treatment, all solutions were prepared in a culture medium suitable for the cell line.

### 2.4. Cell Viability Assay

Cell viability was determined using WST-1 reagent (Roche GmbH, Mannheim, Germany). The reagent is a tetrazolium salt that can be reduced in viable cells to formazan dye by mitochondrial dehydrogenases. The cells were seeded on 96-well microplates (2.5 × 10^3^ cells per well) and left overnight to attach. Then, the culture medium was replaced by the tested compound solutions (1–100 µg/mL) or 1% DMSO solution (control). The cell viability was analyzed after 72 h, followed by incubation for 2 h with WST-1. The Infinite 200 PRO microplate reader (TECAN, Männedorf, Switzerland) was used to read absorbance at 440 nm and 650 nm. The control samples were normalized to 100%, and all tested samples were calculated as the percentage of the control.

### 2.5. Cell Count Analysis

The cell number was assessed by the use of the NucleoCunter NC-3000 image cytometer (ChemoMetec, Lillerød, Denmark) controlled by the NucleoView NC-3000 Software (ChemoMetec, Lillerød, Denmark) according to the manufacturer’s protocol. In brief, cells were seeded at a density of 4 × 10^5^ per T-75 flask and allowed to attach for 24 h. Following treatment, the cells were trypsynized, and samples of the obtained cell suspensions were loaded into the Via1-Cassette (ChemoMetec, Lillerød, Denmark).

### 2.6. Cell Cycle and DNA Fragmentation Assay

The cell cycle phase distribution and DNA fragmentation were assessed using the image cytometer. The analysis was based on differences in the DNA content between the pre-replicative phase (G1 phase) cells, the cells that actually replicate DNA (S phase), the post-replicative plus mitotic (G2-M phase) cells, and the late apoptotic cells. Briefly, cells were seeded at a density of 4 × 10^5^ per T-75 flask and allowed to attach for 24 h. Following treatment, the cells were counted and fixed with ice-cold 70% ethanol. After washing, the cell pellets were stained with solution containing DAPI and Triton X-100 and analyzed using the NC-3000 system according to the protocol “Cell Cycle of Fixed Cells Assay” or “DNA Fragmentation Assay”. Based on the obtained results, the relative ratios of G1/S and G2-M/S were calculated.

### 2.7. Confocal Imaging

Cells were seeded (2 × 10^4^ cells/dish) on separate sterile coverslips placed in Petri dishes and were allowed to attach for 24 h. After treatment, the cells were fixed with 4% paraformaldehyde and permeabilized with 0.1% Triton X-100. Then, the samples were stained with SYTO Deep Red Nucleic Stain (Thermo Fisher Scientific, Waltham, MA, USA) and Phalloidin–Atto 565 (Sigma Aldrich Inc., St. Louis, MO, USA) to visualize the cell nuclei and actin filaments, respectively. The coverslips were mounted onto a microscopic glass slide. The samples were scanned using a Nikon A1R Si confocal imaging system with a Nikon Eclipse Ti-E inverted microscope controlled by Nikon NIS Elements AR software.

### 2.8. In Silico Study

The Absorption, Distribution, Metabolism, Excretion, and Toxicity (ADMET) profiles of the 28-indole-betulin derivatives were predicted in silico using the admetSAR version 2.0 server (East China University of Science and Technology, School of Pharmacy, Shanghai Key Laboratory of New Drug Design, Laboratory of Molecular Modeling and Design; http://lmmd.ecust.edu.cn/admetsar2, accessed on 24 June 2022). This free online tool predicts ADMET profiles based on qualitative classification models built using machine learning methods such as support vector machine (SVM), random forest (RF), and k-nearest neighbors (k-NN), and deep learning methods such as convolutional neural networks (CNN) [33,34]. The calculation of the lipophilicity parameters (MLOGP, WLOGP) was performed by SwissADME (SIB Swiss Institute of Bioinformatics, Molecular Modeling Group, Quartier Sorge, Bâtiment Génopode, Lausanne, CH-1015, Switzerland, http://www.swissadme.ch/index.php, accessed on 16 October 2022) [35].

### 2.9. Statistical Analysis

The data were analyzed using GraphPad Prism 8 (GraphPad Software, San Diego, CA, USA). The statistical significance of the differences was tested by one-way ANOVA with Dunnet’s post hoc test based on the results of three independent experiments. Statistical significance was found with a *p*-value lower than 0.05.

## 3. Results

### 3.1. Chemistry

The compounds EB355A and EB365 were synthesized according to the Steglich method presented in Figure 1. The Steglich method is based on the formation of esters under mild and neutral conditions by using the amide coupling agent *N*,*N*′-dicyclohexylcarbodiimide (DCC) and the organic catalyst 4-dimethylaminopyridine (DMAP). Steglich esterification usually takes place at ambient temperature and neutral pH and produces esters from substrates with sterically hindered substituents, which would undergo elimination under Fischer–Speier esterification conditions [36]. Esterification of the hydroxyl group at the C28 position of betulin and 3-acetylbetulin with 3-indoleacetic acid in the presence of DCC and DMAP in dichloromethane (CH_2_Cl_2_) produced the final products with a 52–58% yield. The 28-indolyl derivatives EB355A and EB365 were purified by column chromatography on silica gel (chloroform/ethanol, 40:1, *v*/*v*). Both compounds were characterized by their ^1^H NMR, ^13^CNMR, and HRMS spectra.

### 3.2. The Effect of Novel Betulin Derivatives on Cancer and Normal Cells Viability

In the first step of our in vitro study, we investigated the effects of EB355A and EB365 on the viability of several human cancer cell lines: A549 (lung carcinoma), MDA-MB-231 (triple-negative breast cancer), MCF-7 (estrogen-receptor-positive breast cancer), DLD-1 and HT-29 (colorectal adenocarcinomas), A375 and C32 (melanomas), as well as normal human fibroblasts. The results from the WST-1 assay are presented in Figure 3. We observed that both the compounds EB365 and EB355A had no impact on the survival rate of normal human fibroblasts. Compound EB365 was found to have weak antitumor activity: in all analyzed cancer cell lines, the viability reduction did not exceed 30% compared to the controls. The MCF-7 cell line was the most sensitive to the triterpene EB365 since the cell survival was about 77% of that of the control following the treatment with the agent in a concentration of 100 µg/mL (156 µM). We revealed that, unlike the EB365 derivative, compound EB355A had a significant anti-cancer effect. It affected the viability of all tested cancer cell lines in a concentration-dependent manner; however, the studied cancer cell lines differed in sensitivity to the derivative. Significant viability reduction (of >40% in comparison to the control) was observed for MCF-7 cells treated with EB355A in concentrations of 50 µg/mL (83 µM) and 100 µg/mL (167 µM), as well as for DLD-1 cells and A375 cells treated with EB355A in a concentration of 100 µg/mL. MCF-7 breast cancer cells and melanoma A375 cells were found to be the most sensitive to the betulin derivative EB355A, since the relative viabilities after incubation with the compound in a concentration of 100 µg/mL (167 µM) were about 20% and 36%, respectively. The estimated EC_50_ values (the concentration that reduced the cell viability by 50% when compared to the untreated control) of EB355A were 40 µg/mL (67 µM), 79 µg/mL (132 µM), and 93 µg/mL (155 µM) for the MCF-7, A375, and DLD-1 cell lines, respectively.

### 3.3. The Effect of EB355A on Proliferation, Cell Cycle, and DNA Fragmentation of MCF-7 Cells

Based on the results of the viability assay, we selected the MCF-7 cell line for subsequent studies aimed at determining whether compound EB355A induced cell cycle arrest and/or cell death. We observed that the total cell number in the population of MC7-7 cells after the treatment with EB355A in concentrations of 50 µg/mL and 100 µg/mL was approximately 2-fold and 3.3-fold lower, respectively, in comparison to the control (Figure 4A). The evaluation of the cell cycle and DNA fragmentation in the breast cancer cells was made using image cytometry. As shown in Figure 4B,D, 100 µg/mL of EB355A caused changes in the cell cycle profile of the tested cell line by increasing the relative ratio of G1/S (5.4 vs. 2.4 in control). Moreover, the performed analysis indicates that the treatment of MCF-7 cells with EB355A resulted in DNA fragmentation. In the cell population exposed to 100 µg/mL of EB355A, the percentage of cells with fragmented DNA was about 54% (Figure 4C,E). Cleavage of cellular DNA into oligonucleosomal fragments is a hallmark of apoptosis. The well-known mechanism of DNA fragmentation during apoptosis is the activation of nucleases, leading to nicks and double-strand breaks within DNA [37,38].

### 3.4. Confocal Imaging of Breast Cancer Cells Treated with EB355A

The anticancer effect of EB355A toward MCF-7 cells was also examined using confocal laser scanning microscopy. The obtained representative microphotographs are compiled in Figure 5. Control cells had a shape characteristic of the cell line and a tendency to aggregate. The incubation with the compound resulted in a growth inhibition manifested by the reduced cell number, and the effect was concentration-dependent. We noticed that MCF-7 cells treated with 100 µg/mL of EB355A cells grew separately or in few-cell groups.

### 3.5. In Silico Analysis of Pharmacokinetic Profile of 28-Indolobetulin Derivatives

A modern approach to the process of developing new drugs is characterized by the fact that at an early stage of research, not only the biological activity of the molecule is determined, but also its pharmacokinetic profile and Absorption, Distribution, Metabolism, Excretion and Toxicity (ADMET) parameters. Problems related to the processes of the absorption, distribution, metabolism, and excretion of molecules with therapeutic potential often lead to unfavorable results at the stage of clinical trials. In the past, the pharmacokinetic profile of promising compounds was mainly determined on the basis of the results obtained in in vivo studies, but now, in vitro and in silico studies are more important in the early phase of development of new bioactive molecules, which can be carried out faster and at a lower cost [39].

Using available online tools, such as the admetSAR version 2.0 server, calculations for the EB355A and EB365 molecules were carried out, the results of which are presented in Table 1.

According to the determined parameters related to the absorption of the drug substance, it can be said that the compounds EB365A and EB366 were characterized by good intestinal absorption (HIA), while they did not show Caco-2 permeability or oral bioavailability. Distribution analysis predicted the location of the tested compounds in the mitochondria and showed their BBB permeability. In addition, the compounds were defined as both P-glycoprotein substrates and inhibitors thereof. The compounds EB365A and EB366 can inhibit the transport of drug substances by acting on transporters such as the organic anion transporting polypeptide (OATP) isoforms 1B1 and 1B3. The studies focused on metabolism have shown that they can be a substrate of the cytochrome P450 3A4 isoform but also an inhibitor of both this isoform and 1A2. The toxicity assessment showed only a moderate probability of toxic effects in the case of both analyzed compounds.

## 4. Discussion

The resistance of cancer cells to available treatment approaches is often a problem in oncology. Thus, novel strategies acting selectively on malignant cells are urgently needed to improve patient outcomes. Many compounds derived from plants are intensively tested for anticancer activity. Previous in vitro and in vivo studies have indicated that betulin, an abundant, naturally occurring triterpene, exerts the desired action against various types of cancer cells by inducing apoptosis and inhibiting cell proliferation [41,42,43]. However, the compound has poor solubility in aqueous media, which is associated with low bioavailability and limited applicability in medicine. It is worth noting that betulin is easily isolated from plant material and the molecule can be modified to obtain more soluble derivatives [44]. In recent years, betulin has been used as a precursor for compounds with promising anticancer properties in vitro [17,45,46,47]. Taking this into account, as well as the fact that the indole nucleus is a biologically accepted pharmacophore in many drugs, including anticancer agents, in this study, we synthesized new 28-indole-betulin derivatives.

In the next step of this work, the obtained compounds were screened for their activity against several cancer cell lines and normal human fibroblasts. Most conventional anticancer chemotherapeutic agents do not have selectivity between normal and cancer cells, which leads to severe side effects and sometimes to the need to discontinue the treatment [48]. In the present study, we demonstrated that the obtained betulin derivatives EB365 and EB355A had no impact on the survival of normal human fibroblasts. Our results indicate that the presence of the acetyl group instead of the free hydroxyl group at the C-3 position of the indole-functionalized derivatives of betulin turned out to be unfavorable for anticancer effects. We found that the cells most sensitive to the anticancer action of EB365 and EB355A betulin derivatives were MCF-7 breast cancer cells. Previously [49], we evaluated an EC_50_ value of cisplatin, the reference antitumor agent, towards MCF-7 cells. The estimated value was 5.5 µM, which is lower than the value demonstrated for EB355A (67 µM). However, as we have shown before [50], cisplatin—in contrast to EB355A—is cytotoxic on normal human fibroblasts (EC_50_ = 9.1 µM). According to our previous study [49], betulin has no impact on MCF-7 cell viability, as observed after 72 h of incubation with the compound in a concentration range of 23–226 µM. Thus, it can be concluded that the presence of the indole group at the C-28 position of the betulin skeleton increases the anticancer potential against the breast cancer cells. Similarly, Güttler et al. [47] synthesized betulin sulfonamides and demonstrated their high antiproliferative and proapoptotic effects in breast cancer cells, suggesting that the betulin derivatives are promising compounds to treat aggressive breast tumors.

Metabolic heterogeneity in cancer cell lines may affect resistance to anticancer therapy. MDA-MB-231 cells, of the triple-negative breast cancer (TNBC) cell line, display low oxidative phosphorylation and high glycolytic activity. In contrast, MCF-7 cells (luminal A breast cancer subtype cell line, estrogen and progesterone receptor-positive, HER2-negative) generate ATP primarily through oxidative phosphorylation [51,52]. In this study, we observed a significantly higher response of the MCF-7 cells than the MDA-MB-231 cells to the treatments with EB365 and EB355A. In our study, the DLD-1 colorectal cancer cell line showed greater sensitivity to EB355A compared to HT-29, which are highly glycolytic colorectal adenocarcinoma cells [53]. We suggest that this may be due to the induction of energy metabolic stress by the indole betulin derivatives. However, this would require further research.

In the present study, the results show that compound EB355A caused the entrapment of MCF-7 cells in the G1 phase, preventing the cells from entering the S phase. Our microscopic observations also revealed significant growth arrest in the population of MCF-7 cells treated with EB355A. Based on our cytometric analysis, we revealed that EB355A induced DNA fragmentation of MCF-7 cells, suggesting that the mode of action of the compound against the luminal breast cancer cells may involve apoptosis induction. This type of programmed cell death is a promising target for anticancer therapy [54]. Our findings support the rationale for further research on EB355A as a potential chemotherapeutic for breast cancer. Moreover, these results indicate that compound EB355A may be used as a valuable skeleton structure for developing novel anticancer agents.

So far, many nitrogen heterocyclic derivatives of betulin and betulinic acid (BA) derivatives have been synthesized, and their potent anti-cancer activities have been demonstrated in in vitro models [30,45,49]. However, many of these compounds do not have the best pharmacokinetic properties or toxicity profile, or there is a lack of relevant data [30]. The structural modification at the C28 position of the triterpene scaffold generally contributes to the enhancement of anti-cancer activity. Yang et al. [55] revealed that several 28-substituted betulinic acid-nitrogen heterocyclic derivatives exert much stronger cytotoxic activity than betulinic acid. In the case of Hela cells, the EC_50_ of their most potent compound containing piperazine moiety was about 2 μM, i.e., 12-fold lower than the value estimated for betulinic acid (EC_50_ = 25 µM). Furthermore, the potential of the derivative to induce apoptosis in Hela cells was also observed. In our previous study, where we analyzed the biological activity of betulin derivatives containing a 1,2,3-triazole ring [49], we demonstrated that the bistriazole of betulin possessing two fluorine moieties displayed a promising EC_50_ value (0.05 μM) against the human ductal carcinoma T47D. Recently, we also found that the 28-substituted derivative of betulone containing 3′-deoxythymidine-5′-yl moiety in the 1,2,3-triazole ring has a significant EC_50_ value (0.17 µM) towards human glioblastoma SNB-19 cells [45]. Interestingly, the above-mentioned nitrogen heterocyclic betulin derivatives differ in activity against particular cancer cell lines. For example, the aforementioned bistriazole derivative of betulin did not show activity against melanoma cells, despite very significant cytotoxicity to the T47D ductal carcinoma cells. In order to explain the different sensitivities of different types of cancer to nitrogen heterocyclic betulin derivatives, it is necessary to investigate the molecular mechanism underlying the anticancer effect.

Among the parameters that may be considered limitations in the optimization of a new molecule, Yang et al. mention such ADMET properties (obtained by in silico methods based on machine learning), such as human intestinal absorption, blood–brain barrier penetration, the inhibition of p-glycoprotein, carcinogenicity, Ames mutagenicity, acute oral toxicity, and CYP450 inhibitory promiscuity [34].

Taking into account the group of parameters related to absorption, the results of the analysis carried out indicate that the tested compounds are characterized by a lack of Caco-2 permeability and human oral bioavailability. These data may suggest routes of administration of the tested compounds other than oral. On the other hand, bioavailability is a complex function of many biological and physicochemical factors. According to the literature data, there is also a relationship between oral bioavailability and intestinal absorption (HIA). As shown in Table 1, both compounds are characterized by a good value of this parameter, which can be treated as an indirect indicator of oral bioavailability [56].

In this respect, the new derivatives do not differ from their precursor, betulin (Appendix A).

Based on computational analysis, mitochondria were predicted to be the subcellular location of the distribution of both EB355A and EB365 molecules, while in the case of betulin, it was predicted to be localized in lysosomes. These data may indicate different metabolic pathways of these compounds.

The blood–brain barrier (BBB) is a physical barrier located between the nervous tissue and blood that protects the brain against the ingress of various xenobiotic substances, such as bacterial toxins, endogenous harmful metabolites, drug molecules, and various particles present in the peripheral blood. The BBB maintains brain homeostasis and functions of the central nervous system [57]. BBB permeability is considered a critical parameter in the ADMET prediction of new molecules [58]. According to the results presented in Table 1, both the compounds EB355A and EB365 have shown the ability to penetrate the BBB, which can be both unfavorable (potential neurotoxicity) and beneficial (e.g., treatment of brain metastasis). Many chemotherapy drugs that are commonly used to treat primary breast cancer are unable to penetrate the blood–brain barrier. The important properties that should be characteristic of drugs capable of penetrating the BBB include the optimal molecular weight in the range of 400–600 Da, higher lipophilicity, and a lower possibility of hydrogen bond formation. In search of new compounds with increased BBB permeability, chemical modifications of drug molecules are introduced by adding lipophilic groups. To determine the effect of introducing the indole system into the betulin molecule, using the SwissADME web tool, theoretical values of the lipophilicity (logPo/w) parameter were determined for the synthesized EB355A and EB365 derivatives, as well as for betulin (as their precursor) and betulinic acid (as a derivative with higher activity described in the literature). It was interesting to determine the values taken into account in the criteria for assessing drug-likeness according to Lipinski (MLOGP) and Ghose (WLOGP) [35]. The corresponding values obtained for betulin (6.0 and 7.0) and betulinic acid (5.82 and 7.09) were significantly lower than for the tested derivatives EB355A (6.61 and 9.27) and EB 365 (6.85 and 9.84) (Appendix A). Lipophilic drugs have a greater tendency to exceed the BBB; however, many of them are cleared by efflux pumps in the family of ATP-binding cassette transporters, P-gp among them. The investigated compounds could be considered potential agents to treat neoplastic changes in the brain resulting from the development of primary breast cancer [59,60]. The in silico analysis performed for betulin predicted that this molecule will not be able to penetrate the blood–brain barrier. This is due to the different structure that does not contain the indole system and, therefore, lower lipophilicity (Appendix A).

The computational algorithms estimated that both tested molecules, EB355A and EB365, can be both substrates and inhibitors of P-glycoprotein and the cytochrome P450 isoform 3A4. Many pharmacokinetic and pharmacodynamic processes in the body are related to transport proteins, including P-glycoprotein belonging to the ABC type transporters (ATP-binding cassette), and with enzymes such as cytochrome P-450 isoenzymes that are responsible for the metabolism of most drugs. The inhibition of these proteins, in the case of using various drugs, may lead to drug–drug interactions. Despite this, drugs that are both substrates and inhibitors of these proteins are used in the treatment of various diseases, including cancer, for example, vincristine, irinotecan, SN-38 (the active metabolite of irinotecan), mitomycin c, and docetaxel [61]. In such a situation, however, attention should be paid to the proper selection of combined drugs, appropriate drug dose adjustments, and monitoring.

However, since P-gp is responsible for the phenomenon of multi-drug resistance and the associated treatment failures in cancer patients, it could be clinically beneficial to block the action of this protein by EB355A or EB365 in cancer cells [62]. It is known that P-glycoprotein expression is detected in a significant percentage of breast cancers [63]. Moreover, P-gp is often overexpressed after exposure to anticancer drugs, leading to a worse response to anti-breast cancer therapy. Thus, combining therapy with anticancer drugs and P-glycoprotein inhibitors seems to be a reasonable strategy. For example, Deshmukh et al. [64] indicated that verapamil, a P-gp inhibitor, sensitizes human breast cancer cells to proteasome inhibitors, enhancing cytotoxic effects and apoptosis.

Apart from ABC proteins, an important role in the transport of many drugs is played by the solute carrier (SLC) protein family. This includes organic anion-transporting polypeptides (OATPs), which have a significant impact on the absorption, distribution, and elimination of many drugs, and their expression is observed in the liver, kidneys, intestine, and the blood–brain barrier. Changing their activity can affect the pharmacokinetic parameters, and thus, the effectiveness of pharmacotherapy [65].

The MATE subfamily (including MATE1 and MATE2), in the kidneys and liver, plays an important role in the efflux of substances consisting mainly of organic cations, but also some anions and zwitterions. OCT proteins are specific membrane transporters in the liver and kidneys. OCT2 proteins facilitate the transfer of cationic compounds from the systemic circulation to the cells of the proximal tubule of the kidney and, together with efflux transporters (mainly MATE1), participate in the excretion of substrates into the urine [66]. Neither of the analyzed compounds, EB355A and EB365, were determined as inhibitors of OCT2, MATE1, or OATP2B1. However, computational algorithms assessed the ability of both molecules to inhibit OATP1B1 and OATP1B3. It can therefore be assumed that these compounds may be potential inhibitors of organic anion-transporting polypeptides. According to the obtained ADMET profile prediction, the tested compounds may be inhibitors of the BSE pump, which is responsible for the transport of bile salts from hepatocytes to the bile ducts. It can be concluded that the betulin derivatives EB355A and EB365 may affect the pharmacokinetic processes, which could be relevant in polypharmacy; however, this does not exclude the potential use of the compounds in vivo.

An important result obtained in the conducted analysis of ADMET is the fact that both molecules were classified into category III [40], i.e., slightly toxic substances. The parameters tested determine the probability of hepatotoxicity at an average level.

## 5. Conclusions

In the current paper, we present a synthesis of new 28-indole-betulin derivatives (EB365 and EB355A). The compounds were evaluated for anticancer activity against human cancer cell lines. We observed significant sensitivity of MCF-7 breast cancer cells to the analyzed betulin derivatives. Particularly active against these cells turned out to be a derivative with a free hydroxyl group at the C3 position, i.e., EB355A, which caused cell growth arrest and DNA fragmentation. Importantly, EB365 and EB355A did not reduce the survival rate of human normal fibroblasts. An in silico ADMET profile analysis of EB355A and EB365 revealed that both compounds are bioactive molecules with relatively low toxicity. Our findings support the rationale for further research on EB355A as a potential chemotherapeutic for luminal breast cancer. Moreover, these results indicate that compound EB355A may be used as a valuable skeleton structure for developing novel anticancer agents.

## Data Availability

Not applicable.

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
