# Peer review of "Synthesis and Anticancer Activity of Indole-Functionalized Derivatives of Betulin"

_pharmaceutics, 2022, doi:10.3390/pharmaceutics14112372_

Round 1

Reviewer 1 Report

The manuscript submitted by Rzepka et al reports the synthesis of two novel indole-betulin analogues that are evaluated for their anticancer activity against different cancer cell lines. This manuscript is well structured overall, and it has been found to add to current research, but it needs some modifications that I have outlined in the comments below.

1.   Please correct the chem draw structures in Scheme 1. Instead of showing O-CO-CH2 show them in bond form. 

2.   The margins of the manuscript are not properly aligned. Please fix them.

3.   In scheme 1: please add the reaction conditions on the arrow (e.g., temperature and reaction time).

4.   The authors have to explain why the product yields i.e., EB355A and EB365 are low. Does the reaction give any side products? if so, did they characterize?

5.   The authors have to include conclusion section at the end.

Reviewer 2 Report

The manuscript by Rzepka and co-workers describes the synthesis, characterization, antiproliferative activity, and in silico analysis of ADME properties of two indole functionalized derivatives of betulin. The authors show the antiproliferative effect of both synthesized compounds against normal human fibroblast and human tumor cell lines A549, MD-MB-231, MCF-7, DLD-1, and HT-29. It is strongly recommended that concentrations be expressed in molar units, including betulin as a control and a first-choice antitumor agent such as cisplatin or doxorubicin as a positive antiproliferative control, just like in previous works of this group. 

The authors report the EC50 of EB355A for MCF-7 and A375 cells; why not for DLD-1 cells?

What can the authors say about the remarkable difference between breast tumor cells MDA-MB-231 and MCF-7?

Figure 3 shows similar behavior for colorectal cancer DLD-1 and HT-29 than the described for breast cancer. Can the authors provide a suggestion for this response? 

How can the structural modifications performed on indole-functionalized betulin help or avoid the antiproliferative response in those specific cells? The authors could provide an in deep discussion considering their previous works with other betulin derivatives. 

How could these modifications change the mechanism of action for betulin/betulinic acid for these new derivatives?

In lines 328-332, the authors state that cytometric analysis revealed apoptosis induction as the cell death pathway induced by betulin derivatives. Can you explain this further?  

How is DNA fragmentation associated with the apoptotic cell death pathway proposed by the authors?

How could ADME properties be related to the differential response in breast and colon cancer?

The obtention of ADME properties of betulin is necessary to establish the advantages/disadvantages of these new derivatives as anticancer agents.

How can be interpreted the CYP inhibitory promiscuity found for both derivatives? (Table 1)

Through the discussion section, the authors describe the expected general response for each of the ADME properties calculated and indicate, based on the values from the table, that betulin derivatives could or could not affect but did not mention how. For example, in lines 349-364 describe the BBB permeability of the compounds and propose them as potential agents to treat neoplastic change in the brain resulting from the development of primary breast cancer. How these compounds could do that?  Due to the substitution, do their lipophilic parameters change substantially compared with betulin or betulinic acid?

Considering the background of the research group and data reported in previous papers, the manuscript needs to improve the discussion and modify the conclusions before being considered for a new peer-review process. 

Reviewer 3 Report

The manuscript presented by Rzepka et al. presents a synthesis and cytotoxic evaluation of betulin derivatives to which an indole scaffold has been added at one of the positions of the starting molecule.

Two new molecules are synthesized in this report. As for the synthesis, there is nothing novel about the design and procedures and the molecules are neither remarkable nor comparable to the examples shown in the introduction of the manuscript. 

In the biological evaluation, cytotoxicity results are presented in different tumor cell lines where one of the compounds, EB355A, is more cytotoxic than the other. Studies in a non-tumor line are also included.

As for the cell cycle assays presented, this reviewer considers that they are not well discussed and the authors should seriously consider the conclusions they draw from them since no arrest in the G2/M phase is observed.

Finally, although the authors present an extensive discussion of their in silico values, the results also add nothing to the manuscript. 

It is considered that this manuscript should not be accepted in the journal Pharmaceutics as it does not report anything new in the field as well as it presents serious flaws in the interpretation of the assays.

Round 2

Reviewer 2 Report

The authors took into account all the recommendations and made the pertinent changes. The manuscript is ready to be published.

Author Response

We wish to thank the Reviewer for the comment.

Reviewer 3 Report

The revised version of the manuscript presented by Rzepka and colleagues has improved the quality of the research presented. However, in the version received by this reviewer, the quality of the images is very low, so for the publication of the article, it should be substantially improved. 

Author Response

We would like to thank the Reviewer for the comment. We assure you that in the original the quality of the figures is good (we have prepared them in high resolution), but when saving them as a pdf, the figures have lost quality. Thus, we have changed the settings in the word document so that there is no automatic compression of images when converting to pdf. We saved the figures in tiff format. We hope the figures' quality is high in the current version of the manuscript, even in the pdf format.